# In Vivo Incorporation of Photoproteins into GroEL Chaperonin Retaining Major Structural and Functional Properties

**DOI:** 10.3390/molecules28041901

**Published:** 2023-02-16

**Authors:** Victor Marchenkov, Tanya Ivashina, Natalia Marchenko, Natalya Ryabova, Olga Selivanova, Alexander Timchenko, Hiroshi Kihara, Vladimir Ksenzenko, Gennady Semisotnov

**Affiliations:** 1Institute of Protein Research, Russian Academy of Sciences, 4 Institutskaya St., 142290 Pushchino, Russia; 2Skryabin Institute of Biochemistry and Physiology of Microorganisms, Russian Academy of Sciences, 5 Prospect Nauki, 142290 Pushchino, Russia; 3Department of Physics, Kansai Medical University, Shin-Machi 2-5-1, Hirakata 573-1010, Japan

**Keywords:** GroEL chaperonin, photoproteins, protein–protein interactions, protein oligomerization, cell division

## Abstract

The incorporation of photoproteins into proteins of interest allows the study of either their localization or intermolecular interactions in the cell. Here we demonstrate the possibility of in vivo incorporating the photoprotein *Aequorea victoria* enhanced green fluorescent protein (EGFP) or *Gaussia princeps* luciferase (GLuc) into the tetradecameric quaternary structure of GroEL chaperonin and describe some physicochemical properties of the labeled chaperonin. Using size-exclusion and affinity chromatography, electrophoresis, fluorescent and electron transmission microscopy (ETM), small-angle X-ray scattering (SAXS), and bioluminescence resonance energy transfer (BRET), we show the following: (i) The GroEL_14_-EGFP is evenly distributed within normally divided *E. coli* cells, while gigantic undivided cells are characterized by the uneven distribution of the labeled GroEL_14_ which is mainly localized close to the cellular periplasm; (ii) EGFP and likely GLuc are located within the inner cavity of one of the two GroEL chaperonin rings and do not essentially influence the protein oligomeric structure; (iii) GroEL_14_ containing either EGFP or GLuc is capable of interacting with non-native proteins and the cochaperonin GroES.

## 1. Introduction

Chaperonins are oligomeric ring-shaped proteins forming seven- to nine-fold symmetric one- or double-ring complexes of 500–1000 kDa, resulting in one or two extensive inner cavities [1,2,3]. They are widely distributed in various organisms and play an essential role in the proteostasis network, including the facilitation of protein folding and transport and the prevention of protein misfolding and aggregation [1,4,5,6,7]. Despite the relatively low sequence identity between chaperonins from different organisms, their spatial structures are very similar [2,8,9]. Each subunit of a chaperonin consists of three domains: apical, intermediate, and equatorial [8,9,10], whose allosteric conformational changes are induced by Mg-ATP binding, while its hydrolysis is crucial for chaperonin functioning [11,12,13,14].

On the top of apical domains, there are hydrophobic clusters responsible for binding to various hydrophobic species of both high (denatured or non-native proteins) and low molecular weight (various peptides and drugs [8,15,16,17,18]). Equatorial domains contain adenyl nucleotide binding sites and provide the majority of interactions both between the subunits and between the rings [8,13,14]. The protein C- and N-ends are located close to each other within equatorial domains in the protein’s inner cavity [8]. The intermediate domains act as hinges between apical and equatorial domains, providing allosteric conformational changes [8,13,19]. The functional and structural peculiarities of chaperonins are of importance due to their potential usage in biotechnology and biomedicine projects [18,20,21,22,23]. Some investigators used fusions within the GroEL cavity to produce toxic or prone-to-aggregation polypeptides [21,22], while others employed the tight binding of hydrophobic drugs by GroEL apical domains for the delivery of these drugs to tumors [18]. Works on the direct incorporation of polypeptides into GroEL-like chaperonins are not numerous. For example, in the work of Kyratsous et al. [24], two proteins, which are normally insoluble during bacterial expression, mouse prion protein PrP and Varicella Zoster virus protein ORF21p, were used as a fusion part of the GroEL subunit. The authors used a vector where the sequence of a target polypeptide is placed in frame at the carboxy terminus of the GroEL subunit and demonstrated that these fusion constructs are expressed in large amounts in a soluble form [24]. Similarly, usually insoluble *Shigella*’s IpaB antigen domain was fused with GroEL from *Salmonella enterica* Typhi bacteria to receive a fusion product for the immunization of mice [25]. Unfortunately, no analysis of the location of fused target proteins within GroEL oligomeric structure was made in these works, and the authors explained the effect of increased solubility by an increase in the chaperonin local concentration near the folding target protein. However, a set of bacterial expression vectors has been published that allows fusing the target protein with GroEL for its subsequent expression in a soluble, easily purified form [26]. A more thoughtful strategy was used by Fedorov’s group to introduce in GroEL some toxic-to-cell and prone-to-aggregation polypeptides. The peptide was inserted into the GroEL polypeptide chain in such a way that it did not interfere with the formation of the correct GroEL oligomeric structure, but protruded into the oligomer cavity and interacted with the substrate-binding surface. For this purpose, a methionineless variant of thermostable chaperon GroEL from *Thermus thermophilus* was designed [27], and two target peptides—a toxic antimicrobial polypeptide polyphemusin I [22] and a therapeutic peptide for the treatment of AIDS patients enfuvirtide [28]—were used. In both cases, the polypeptide fused with GroEL was expressed with high yield in a soluble state, and the target polypeptide was separated from GroEL by CNBr cleavage. It should be noted that in the case of enfuvirtide, the fusion protein was expressed as a monomer and not as an oligomer. The authors attribute this to the effect of electrostatic repulsion [22,23]. Another interesting approach was used to incorporate a target protein into the oligomeric double-ring structure of *Thermococcus* sp. KS-1 chaperonin α-subunit (TCP) [21]. Expression plasmids carried four TCP genes ligated head to tail in phase and a target protein gene at the 3′ end of the linked TCP genes. As a reporter test, protein GFP was used. The final fusion protein was double-ring oligomeric, possessed green fluorescence, and electron microscopy image analyses indicated that the GFP was accommodated in the central cavity. Using this strategy, TCP fusions with two virus proteins toxic to host cells or two antibody fragments prone to aggregation were well expressed as double-ring complexes in the soluble fraction of *E. coli* [21].

Still, the incorporation of a protein of interest into the chaperonin inner cavity is tricky and requires a special approach. It depends on the protein size and the volume of the chaperonin inner cavity [21]. Moreover, a bulky protein fused with the chaperonin subunit hampers the formation of the chaperonin native oligomeric structure. To overcome this difficulty, a special design and technology of protein incorporation into the chaperonin cavity have been developed [21]. Here we propose a simple method of incorporation of photoproteins (and likely other proteins) of appropriate size fused with the GroEL subunit C-end into the inner cavity of tetradecameric GroEL chaperonin.

The photoproteins used in this study are the enhanced green fluorescent protein (EGFP), 27 kDa, and *Gaussia princeps* luciferase (GLuc), 20 kDa. Apart from being model proteins, these luminescent labels can show the chaperonin localization within the cell and its intermolecular interactions [29]. The constructed plasmids encoding the fusion proteins GroEL-EGFP and GroEL-GLuc were expressed in *Escherichia coli*. The GroEL-EGFP fusion proved the most informative. The expression of the plasmid encoding GroEL-EGFP alone does not result in the formation of a tetradecameric double-ring GroEL particle; the yielded product is seemingly monomeric GroEL-EGFP fusion with a molecular weight of about 90 kDa, as evaluated by size-exclusion chromatography. In contrast, when coexpressed with the plasmid producing excessive wild-type GroEL subunits, this plasmid provides the full-size GroEL_14_ particles with incorporated EGFP. Fluorescent microscopy shows that the GroEL_14_-EGFP particles are evenly distributed inside the normally divided cells, while within abnormal gigantic nondivisible cells, they are concentrated mostly near the periplasm. The product of this coexpression was purified and analyzed by polyacrylamide gel electrophoresis (PAGE), fluorescence, absorption, transmission electron microscopy (TEM), and small-angle X-ray scattering (SAXS). It contains full-size GroEL_14_ particles with incorporated EGFP (GroEL_14_-EGFP particles). The amount of incorporated EGFP was estimated to be 1 per 12 ± 2 wild-type (WT) GroEL subunits, i.e., 1 subunit fused with EGFP in a tetradecameric GroEL_14_ particle. The SAXS data and structural reconstitution of TEM images suggest the EGFP localization inside the GroEL inner cavity. The presence of a fluorescent protein in the tetradecameric GroEL_14_ population allows for tracing its behavior by EGFP fluorescence and leads to the conclusion that tetradecameric GroEL_14_-EGFP can interact with GroES and non-native proteins.

Using the same technique of coexpression in *E. coli* of two plasmids encoding the fusion GroEL-GLuc and the WT GroEL subunit, we obtained tetradecameric GroEL_14_ with incorporated GLuc (GroEL_14_-GLuc) and analyzed its binding to GroES and non-native proteins in both cell lysate and purified samples. Using BRET, the distance between the fluorescein-labeled non-native protein and the bioluminescent chromophore of GLuc incorporated into GroEL_14_ was estimated as ~70 Å, which hints at the localization of the non-native protein bound mainly to the GLuc-free end of the GroEL_14_-GLuc cylinder.

## 2. Results

### 2.1. The Analysis of Plasmid Expression Products

The basic condition for the successful in vivo incorporation of photoproteins into GroEL_14_ tetradecameric quaternary structure is the coexpression of two plasmids. One of them encodes the WT GroEL subunit and possesses a high copyability while the other encodes the fusion GroEL-EGFP and displays a much lower copyability (see Materials and Methods (Section 4)). A schematic representation of these genetic constructs is shown in Figure 1a,b.

Figure 1c,d shows fluorescent microscopy images of *E. coli* cells after the expression of pAC28 groEL-EGFP alone and its coexpression with pET11c groES/EL. Size-exclusion chromatography profiles of the corresponding soluble fractions of cell lysate are presented in Figure 1e,f. These data are evidence for the following. Firstly, the folding of EGFP as a part of GroEL-EGFP results in its physiological intrinsic green fluorescence. Secondly, there is an even diffuse distribution of GroEL-EGFP within normally divided cells and uneven distribution within abnormal nondivisible cells where no fluorescence is observed within the central part of the gigantic cells. Thirdly, the expression of pAC28 groEL-EGFP encoding GroEL-EGFP alone results in producing mostly monomeric GroEL_1_-EGFP fusion (~90 kDa), whereas the product of its coexpression with pET11c groES/EL encoding the groE operon of wild type is the tetradecameric double-ring GroEL_14_-EGFP (~800 kDa). Note that nondivisible gigantic cells are observed only in the case of the coexpression, i.e., when full-size tetradecameric GroEL_14_ particles are overproduced (Figure 1c,d).

The populations of GroEL_14_-EGFP and GroEL_1_-EGFP were purified (see Materials and Methods (Section 4)) to reveal some of their physicochemical properties. First, the amount of GroEL_1_-EGFP fusion in the population of full-size tetradecameric GroEL_14_-EGFP was estimated using SDS-PAGE and spectrophotometry (Figure 2). It appeared to be 1 fusion GroEL-EGFP subunit per 12 ± 2 WT subunits, as follows from the intensity of electrophoretic bands corresponding to the Coomassie-stained WT GroEL (~60 kDa) and GroEL-EGFP (~90 kDa) (Figure 2a) and from the spectrophotometric analysis (Figure 2b).

### 2.2. Physicochemical Properties of GroEL_14_ with Incorporated EGFP

Figure 3 shows SAXS patterns and Guinier plots of WT GroEL_14_ and GroEL_14_-EGFP populations purified after expression of pET11c groES/EL alone and with pAC28 GroELgroEL-EGFP. Although these patterns are very similar, which implies the similarity of the protein quaternary structures, there are two distinctions. One is a more pronounced minimum in the SAXS pattern at 0.055 (Å^−1^) of the population with EGFP incorporated. The other is a decrease in the radius of gyration (Rg) by 5 Å for the GroEL_14_-EGFP as compared to GroEL_14_. The construction of the SAXS pattern in the form of a Guinier plot allows us to evaluate the radius of gyration and molecular mass of the proteins studied [30]. The radius of gyration is determined using the tangent of the slant of the linear part of the Guinier plot [30]. In our case, the linear part of the Guinier plot is within the scattering vector (*h*) range of 0.009 to 0.018 Å^−1^ (Figure 3b,d). The extrapolation of the Guinier plot to zero of the scattering vector (*h*) allows determining the scattering intensity at the zero-scattering angle. This value contains the information about the molecular mass of the proteins studied, if the correction for protein concentration has been made, and if some standard with known molecular mass was used [30]. It is interesting that after correction on the protein concentration, the I(0) values for both populations are close to each other, hinting at the proximity of the molecular weights of these proteins. The other possibility to estimate independently the Guinier parameters and molecular weight of the proteins studied is represented in the recent works of Piiadov et al. [31,32]. We uploaded our SAXS data to the online calculator described in these works and obtained the following results. SAXS pattern of WT GroEL_14_ population: Guinier limits: from 0.007 Å^−1^ to 0.019 Å^−1^; Rg = 68 Å; integration limit *h**Rg = 1.300 (using 8/Rg); found molecular weight: 1400 kDa. SAXS pattern of GroEL_14_-EGFP population: Guinier limits: from 0.007 Å^−1^ to 0.020 Å^−1^; Rg = 63 Å; integration limit *h**Rg = 1.286 (using 8/Rg); found molecular weight: 974 kDa. The Rg decrease may be caused by both the EGFP incorporation into the GroEL_14_ inner cavity because it reflects an increase in the protein density arising with proximity to the protein center of mass and the decrease in the protein hydrodynamic size. Unfortunately, the molecular weight of the protein studied is closely related to the radius of gyration in the approach proposed by Piiadov et al. [31,32], and the reason for the Rg value change is not taken into account. Nevertheless, within the error of the method [31], the molecular weight estimated seems to be real for GroEL_14_. The SAXS pattern of GroEL at 0.055 (Å^−1^) of the scattering vector is mainly sensitive to an interring distance (our unpublished data).

The transmission electron microscopy fields of vision of both products are shown in Figure 4. One looks like the known side and top images of the WT GroEL_14_ particles [8,13,33] (Figure 4a,c). The images of GroEL_14_-EGFP (Figure 4b,d) are different from those of the WT GroEL_14_ due to a certain electron density inside the GroEL_14_ inner cavity and perturbation in the region of the protein equatorial domains resulting in the absence of some interring distance which is clearly visible in WT GroEL_14_ side view. The structural reconstructions of the images by the program EMAN2 [34] (see Materials and Methods (Section 4)) (Figure 4e,f) confirmed the presence of the EGFP fused part within the inner cavity of one ring of the tetradecameric GroEL_14_ double-ring particle and decrease the interring distance in the case of the GroEL_14_-EGFP population. The resolution of the models of GroEL_14_ and GroEL_14_-EGFP obtained after refinement was 20–21 Å and 22–26 Å correspondingly.

To verify the ability of GroEL_14_-EGFP to bind to GroES and denatured proteins, we used nondenaturing PAGE and affinity chromatography based on denatured lysozyme (see [35]). Figure 5a presents the nondenaturing PAGE of the coexpression product GroEL_14_-EGFP without and with GroES and Mg-ADP and shows protein electrophoretic bands detected by Coomassie staining and fluorescence (see Materials and Methods (Section 4)). As seen, in the presence of GroES and Mg-ADP, the mobility of the part of the protein band is noticeably retarded in the case of both Coomassie staining and fluorescence (Figure 5a). It means that GroEL_14_-EGFP can bind to GroES. Moreover, GroEL_14_-EGFP can tightly interact with denatured lysozyme (Figure 5b). Thus, the GroEL_14_ particles with EGFP incorporated in their inner cavity retain the main chaperonin functions, i.e., the ability to bind non-native proteins and GroES.

The same technology, i.e., coexpression of two plasmids encoding the GroEL subunit fused with photoprotein and the wild-type subunit, was applied to incorporate another photoprotein, luciferase of *Gaussia princeps*, into the GroEL_14_ particle.

### 2.3. Physicochemical Properties of the GroEL_14_ Particle with Incorporated GLuc

Figure 6 shows the bioluminescence spectrum of *E. coli* cells after coexpression of the plasmids encoding GroEL fused with GLuc and WT GroEL (Figure 6a), and nondenaturing PAGE of soluble fractions of their lysates (Figure 6b) (see Materials and Methods (Section 4)). The bioluminescence spectrum (Figure 6a) confirms that the luciferase is folded into a physiologically active conformation after biosynthesis in the cell as fused with the GroEL subunit. The electrophoretic mobility of the bioluminescent protein bands corresponds either to the mobility of full-size GroEL_14_-GLuc or to that of the complex of GroEL_14_-GLuc with endogenous GroES (Figure 6b).

To study the interaction of GroEL_14_-GLuc with denatured proteins, the coexpression product was purified using the standard technique for GroEL purification [33,36]. Figure 7 shows the GroEL_14_-GLuc bioluminescence spectra in the absence and presence of some fluorescein-labeled denatured proteins (see Materials and Methods (Section 4)). As seen, in the presence of fluorescein-labeled non-native proteins, the bioluminescence spectrum contains two components: bioluminescence of GroEL_14_-GLuc and emission of the fluorescein label attached to the denatured protein. The fluorescein emission component is likely induced by the bioluminescence resonance energy transfer from GroEL_14_-GLuc to the fluorescein-labeled denatured protein. The distance between the donor (GLuc) and acceptor (fluorescein) chromophores was estimated as 70 ± 2 Å, according to our method developed previously [37]. Thus, the full-size GroEL_14_ particles with incorporated EGFP or GLuc can bind GroES and denatured proteins, thereby displaying the intrinsic chaperonin properties.

## 3. Discussion

In this work, the key point is the proposed simple technique of incorporation of a photoprotein or any other protein of interest into the internal cavity of GroEL without significantly affecting its oligomeric structure and main functional properties, i.e., the ability to bind the cochaperonin GroES and non-native proteins. This technique is based on the coexpression of two plasmids. One of them encodes the GroEL subunit fused with a protein of interest of appropriate size to fit the GroEL inner cavity (≤30 kDa), while the other encodes WT GroEL. These genetic constructs provide the excess of WT GroEL over that of the GroEL-fused protein of interest during the GroEL_14_ assembly in vivo, which allows the spontaneous incorporation of a GroEL-fused protein molecule into the chaperonin inner cavity formed by the WT subunits. A disadvantage of this technique is the inability to incorporate more than one alien protein molecule into the tetradecameric GroEL_14_ particle, i.e., only one ring of GroEL_14_ can be occupied with an alien protein (Figure 2 and Figure 4). GroEL crystal structure allows fitting EGFP into the inner cavity of the chaperonin without essential disruption of its quaternary structure (Figure 8). Nevertheless, it draws attention to the reduction of the interring distance that follows from the structural reconstruction of TEM images (Figure 4).

The reason for such selectivity requires further research. In principle, the yield of the GroEL subunit fused with a photoprotein can be increased by using genetic constructs with promoters controlled by, e.g., arabinose, to adjust the biosynthesis level of GroEL fused with the protein of interest to that of WT GroEL. Nevertheless, the distance between GLuc and fluorescein-labeled denatured proteins was estimated from BRET to be 70 Å ± 2 Å. This hints that the binding of denatured proteins occurs mainly at the ring opposite to the one occupied by GLuc (Figure 7). It is possible that the protein located in the inner cavity of GroEL creates some steric restrictions on the binding of denatured protein in this cavity. The fluorescence microscopy of *E. coli* cells after coexpression of two plasmids, one encoding WT GroEL and the other for EGFP fusion, shows the uneven distribution of GroEL_14_-EGFP within nondivisible cells, while in normally divided cells, its distribution is diffused (Figure 1). This phenomenon is of interest and requires a special study. It is possibly caused by the interaction of oversynthesized full-size GroEL_14_-EGFP with cell division proteins, e.g., FtsZ [41], and inhibition of cell division. The presence of EGFP in GroEL provides information about the size of molecules directly in the cell lysate without their purification (Figure 1). GroEL_14_-EGFP particles have a decreased radius of gyration as compared to GroEL_14_ particles (Figure 3), thereby suggesting EGFP localization inside the GroEL_14_ cavity and the decrease in the protein interring distance. The structural reconstruction of GroEL_14_ and GroEL_14_-EGFP using relevant electron microscopy images supports this conclusion. GroEL_14_-EGFP was tested for the ability to interact with the cochaperonin GroES and a denatured protein, in our case, lysozyme lacking disulfide bonds [35]. The test confirmed that the GroEL_14_-EGFP particle retains its main functional features—the interaction with GroES and the denatured protein. Using the same technique of coexpression, the smaller photoprotein, *Gaussia princeps* luciferase of about 20 kDa, was apparently incorporated into the inner cavity of one GroEL ring. The protein GroEL_14_-GLuc exhibits specific GLuc bioluminescence and interacts with endogenous GroES from the cell lysate, as shown by nondenaturing electrophoresis, followed by bioluminescence recording of coelenterazine-treated gel, which is very useful in studying heterogeneous solutions. Thus, our finding is that the incorporation of proteins of appropriate size into the GroEL_14_ inner cavity can spontaneously occur in vivo resulting from the coexpression of two plasmids one of which encodes the GroEL subunit fused with the protein of interest while the other encodes WT GroEL, with the excess of the latter over the former. This technique can be used in biotechnology and biomedicine.

## 4. Materials and Methods

### 4.1. Chemicals and Related Enzymes

In the present work, the following reagents were used: HCl, NaOH, NaCl (Reachem, Moscow, Russia); acrylamide, β-mercaptoethanol, molecular weight protein markers, ammonium sulfate, SDS (Sigma-Aldrich, St. Louis, MO, USA); buffer components HEPES, KCl, DTT, MgCl_2_, CaCl_2_, Na_2_HPO_4_, NaH_2_PO_4_ (ICN Biomedicals, Costa Mesa, CA, USA); coelenterazine (Intrinsic Bioprobes, Tempe, AZ, USA). Substrate proteins for GroEL_14_: lysozyme, α-lactalbumin, and malate dehydrogenase were purchased from Sigma-Aldrich (St. Louis, MO, USA).

### 4.2. Bacterial Strains, Culture Conditions, and Enzymes

*Escherichia coli* strain DH5α (Novagen, Madison, WI, USA) was used for DNA plasmid propagation, and *E*. *coli* strain BL21(DE3) (Novagen, Madison, WI, USA) was used for gene expression. *E. coli* cells were routinely grown in lysogeny broth (LB) [42] at 37 °C. For the selection and maintenance of the plasmids, LB medium was supplemented with ampicillin at 100 μg mL^−1^ and kanamycin at 20 μg mL^−1^. Restriction enzymes and nucleases, T4 ligase, dNTP_S_, and isopropyl β-D-1-thiogalactopyranoside (IPTG) were from Thermo Fisher Scientific (Vilnius, Lithuania). Pfu Turbo high-fidelity DNA polymerase was from Alfa Ferment (Moscow, Russia). All enzymes were used as recommended by the suppliers.

### 4.3. Genetic Manipulations

Standard procedures were used for DNA restriction, ligation, gel electrophoresis, and transformation of *E. coli* [42]. Plasmid DNA was purified using QIAprep Spin Miniprep Kit (Qiagen, Hilden, Germany). QIAquick gel extraction and QIAquick PCR purification kits (Qiagen, Hilden, Germany) were used for DNA isolation from agarose gels and purification of PCR products.

To construct fused in frame *groEL* with EGFP or luciferase GLuc from *Gaussia princeps* genes, the low copy (~ 10 copies per cell) expression plasmid pAC28 [43] was used. The *groEL* without stop codon was amplified with the pair of primers 5′-TTCCATGGCAGCTAAAGACGTAAAATTC-3′ and 5′-TTGAATTCCGGCATCATGCCGCCCATG-3′ (restriction sites are underlined) and cloned into NcoI/EcoRI sites of pAC28 to give pAC28 groEL. The EGFP was amplified using the 5′-TTGAATTCGGTGTGAGCAAGGGCGAGGAG-3′ and 5′-TTGTCGACTTACTTGTACAGCTCGTCCATGC-3′ primers. The DNA fragment containing the GLuc gene without the 17 a.a. signal peptide was amplified with the 5′-TTGAATTCGGGAAACCAACTGAAAACAATGAAGAT-3′ and 5′-TTGTCGACTTAATCACCACCGGCACCCT-3′ primers. The PCR products were cloned independently into the pAC28 groEL plasmid at the EcoRI/SalI sites. The resultant plasmids allow producing fused proteins with a tetra-peptide linker, PEFG, between GroEL and EGFP or GLuc.

The recombinant plasmid containing *groES-groEL* operon was constructed as follows. The DNA fragment containing *groES-groEL* operon was amplified using the 5′-TTATTTCATATGAATATTCGTCCATTGCATG-3′ and 5′-TTGTCGACTTACATCATGCCGCCCAT-3 primers, respectively, and pACYC184 *groES-groEL* (gift of A.S. Girshovich [33]) as a template. The resulting amplicon was cloned into the NdeI/SalI sites of medium copy number pET11cjoe vector (gift of H.J. Khackmuss, Stuttgart University, Stuttgart, Germany) to create pET *groES-groEL* compatible with pAC28. The constructed plasmids were employed for the coexpression in *E. coli* BL21(DE3) cells.

### 4.4. Protein Purification and Labeling

GroEL, both of wild type and with photoproteins incorporated, as well as GroES, were isolated using standard published protocols [33].

To prepare the proteins labeled with fluorescein, their 5–10 mg/mL solutions in 0.2 M NaHCO_3_, pH 8.4, were incubated with a 5-fold molar excess of 5- (and 6)-carboxyfluorescein succinimidyl ester “mixed isomers” (Invitrogen, Waltham, MA, USA) at intensive mixing with Heidolph Reax top (Heidolph Instruments, Schwabach, Germany) for 1 h at room temperature. The reaction was stopped by the addition of 1 M Tris-HCl, pH 7.6, up to 10 mM. Nonbound labels were removed using a G25 gel chromatographic column PD10. The labeled proteins were fractionated additionally with DEAE-Toyo Pearl (HαLA) or CM-Toyo Pearl (lysozyme and MDH) to separate the populations with different amounts of the labels. After that, each fraction underwent gel filtration with a Sephacryl S100 1.6 × 60 cm column in 20 mM NaHCO_3_, pH 8.4, to achieve the most homogeneous labeled protein. The concentration of fluorescein-labeled proteins was determined spectrophotometrically after subtracting the contribution of the fluorescent label (using the known fluorescein extinction coefficient at 490 nm of 87,000 M^−1^ cm^−1^) and confirmed by SDS-PAGE densitometric analysis. The denatured states of lysozyme and α-lactalbumin were attained by reducing intermolecular disulfide bonds with DTT [35], while that of MDH was by dilution from 8 M urea solution.

### 4.5. Registration of Fluorescence and Bioluminescence in the Cells, Cell Extracts, Electrophoretic Gels, and Pure Protein Solutions

The aliquots of precipitate cells were resuspended in 80 μL of 20 mM Tris-HCl, pH 7.6, 150 mM NaCl up to OD_590nm_ = 0.25. To measure the bioluminescence spectra, 25 μM coelenterazine (CZ) was added to the cell suspension at mixing. Using a quartz cuvette 3 × 3 × 5 mm, 10 bioluminescence spectra of the solution were recorded with a Varian Cary Eclipse spectrofluorimeter (Palo Alto, CA, USA) in the mode “the lamp is off” within the wavelength region from 330 nm up to 730 nm. The spectra were corrected to the time-dependent decay of bioluminescence occurring due to coelenterazine consumption, its penetration into cells, etc., using the bioluminescence decrease at 470 nm and then averaged, followed by the resultant value correction for spectral sensitivity of the device [44]. Note that the time-dependent bioluminescence decrease in the case of *Gaussia princeps* luciferase is not significant and occurs within 300 s by only 25%. The EGFP fluorescence spectra were recorded with excitation at 470 nm.

The same procedures were applied to record luminescence of cell lysates and purified proteins. To register EGFP fluorescence in the gel after nondenaturing PAGE of GroEL_14_-EGFP fusions, the electrophoretic plates with the gel inside were photographed through an orange glass filter GS-20 (Krasnyj Gigant, Nikolsk, Russia) with the maximum transmission at the green region of the spectrum while illuminating the bottom and top with a group of blue LED_S_ arranged in a line. To register GLuc bioluminescence, one of two glass plates of electrophoretic gel was removed, and on the open side of the gel, 1 mL of 20 μM coelenterazine solution in 20 mM Tris-HCl, 100 mM KCl, and 10 mM MgCl_2_ was evenly loaded. The plate was placed in a darkened box and photographed with a long exposure of 30 s.

The distances between fluorescein and GLuc chromophores attached to proteins were evaluated from BRET according to the published method [37].

The protein absorption spectra were recorded using a Cary 100 Bio spectrophotometer (Varian Medical Systems, Palo Alto, CA, USA). Protein concentrations were determined using molar extinction coefficients at 280 nm of 9080 M^−1^ cm^−1^ for WT GroEL_14_ [45], at 280 nm of 1280 M^−1^ cm^−1^ for WT GroES_7_ [45], and at 280 nm of 22,000 M^−1^ cm^−1^ and at 489 nm of 56,000 M^−1^ cm^−1^ for EGFP [46].

The size-exclusion chromatography was performed using a ProStar HPLC chromatograph (Varian Medical Systems, Palo Alto, CA, USA), a Superose 6 10/300 GL column, and a flow rate of 0.4 mL/min; the column was calibrated using the following proteins (from Gel Filtration Calibration Kit, Pharmacia, Uppsala, Sweden; Sigma-Aldrich, St. Louis, MO, USA): WT GroEL_14_ (800 kDa), thyroglobulin (669 kDa), bovine serum albumin trimer (198 kDa), bovine serum albumin monomer (66 kDa), catalase monomer (60 kDa), ovalbumin (45 kDa), cytochrome *c* (12.3 kDa), insulin (7 kDa). The chromatographic profiles were simultaneously recorded by EGFP fluorescence with λ_ex_ = 280 nm, λ_em_ = 515 nm, and absorbance at 280 nm.

### 4.6. SAXS Measurements

X-ray solution scattering measurements were performed on the beam line BL-6A small-angle installation. A stable beam of photons with a wavelength of 1.503 Å was provided by a bent-crystal horizontally focusing monochromator and a vertically focusing mirror [47] of the Photon Factory, National Laboratory for High Energy Physics, Tsukuba, Japan. A camera with a 2.35 m sample-to-detector distance collected data in the range of the scattering vector (*h*) from 0.008 to 0.2 Å^−1^. The background data for the buffer solvent were collected before or after data collection for the protein solution. The temperature was kept at 23 °C for all the measurements. The data were registered by the two-dimensional CCD X-ray detector PILATUS 100 K. For SAXS data integration and Guinier analysis ATSAS online 3.2.1 software (BioSAXS group, EMBL Hamburg, Germany) was used.

### 4.7. Transmission Electron Microscopy and Images Processing

Samples for electron microscopy (EM) studies were prepared according to the negative staining method. A copper 400-mesh grid (Electron Microscopy Science, Hatfield, PA, USA) coated with a formvar film (0.2% formvar solution in chloroform) was mounted on a sample drop (10 μL). After 5 min absorption, the grid with the preparation was negatively stained for 1.5–2.0 min with 1% (*w*/*v*) aqueous solution of uranyl acetate. The excess of the staining agent was removed with filter paper. The preparations were analyzed using a JEM-1200EX transmission electron microscope (JEOL, Ltd., Tokyo, Japan) at the accelerating voltage of 80 kV. Images were recorded on the Kodak electron image film (SO-163) at nominal magnification of 40,000.

Reference-free 2D classification, initial model building, 3D classification, 3D refinement, and CTF refinement were performed using the EMAN2.22 software package (Baylor College of Medicine, Houston, TX, USA) [34]. The EMAN2.22 software package is a broadly based greyscale scientific image processing suite with a primary focus on processing data from transmission electron microscopes. Despite EMAN’s original purpose was performing single particle reconstructions (3D volumetric models from 2D cryo-EM images) in this work we used this program for processing data from transmission electron microscopy images of negatively stained molecules WT GroEL_14_ and GroEL_14_-EGFP. Particles were manually picked by the e2boxer.py program from the software package EMAN2. A total number of 1262 particles for GroEL and 1586 particles for GroEL-EGFP were used to generate 32 class averages and to build the final 3D reconstruction. At modeling, we used only one limitation—conservation of C7 or D7 symmetry, i.e., both rings are identical. In the case of the removal of this limitation, the models were very distorted and wry.

### 4.8. Fluorescence Microscopy

To obtain the fluorescent microscopy images the sterile glass plates were coated with 1% agarose in deionized water and dried in ultraviolet light. Aliquots (5–10 µL) of the *E. coli* cells transformed by the recombinant plasmids were loaded on these plates, covered with cover glasses and immediately visualized with Leica TCS SP5 Confocal Microscopy System (Leica Microsystems, Wetzlar, Germany). The excitation wavelength of the argon laser was 488 nm, and luminescence was measured in the EGFP luminescence band (500–550 nm). The images were analyzed using the Leica Application Suite AF 2.6.3 program (Leica Microsystems, Wetzlar, Germany).

## Figures and Tables

**Figure 1 molecules-28-01901-f001:**
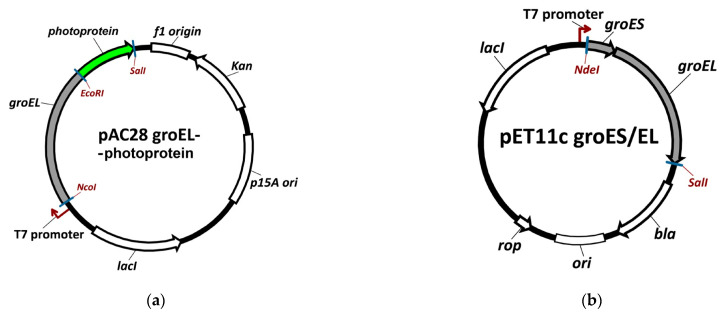
Incorporation of the photoprotein EGFP into the GroEL_14_ structure: schematic representation of the genetic constructs encoding the GroEL subunit fused with photoprotein (**a**) and the WT groE operon (GroEL + GroES genes) of *E. coli* cells (**b**); fluorescent microscopy images of *E. coli* cells after the expression of pAC28 groEL-EGFP alone (**c**) and its coexpression with pET11c groES/EL (**d**); size-exclusion chromatography profiles of the corresponding cell lysates are shown in (**e**,**f**), and the column calibration is in (**g**); the green arrows indicate the elution times of monomeric GroEL_1_-EGFP and oligomeric GroEL_14_-EGFP. Abbreviations in the schemes (**a**,**b**) mean the following: pAC28 groEL-photoprotein and pET11c groES/EL are names of the constructed plasmids; *Kan* is the gene encoding resistance to kanamycin; *p15A ori* is the origin of replication p15A; *lacI* is the *lac* repressor; *bla* is the beta-lactamase gene; *ori* is the origin of replication ColE1; *rop* is the gene of Rop protein; *NcoI*, *EcoRI*, *SalI*, and *NdeI* are restriction sites.

**Figure 2 molecules-28-01901-f002:**
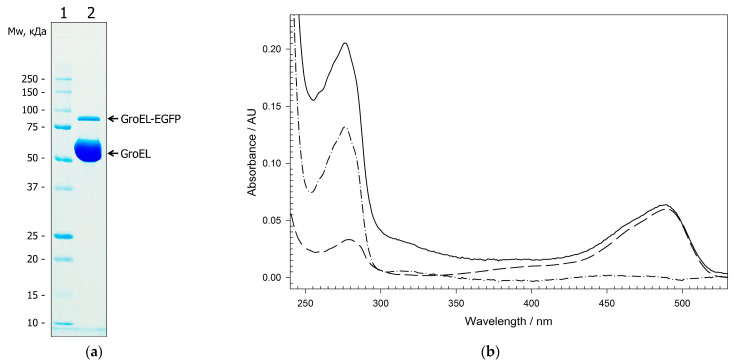
An estimation of the amount of fusion protein GroEL_1_-EGFP incorporated in tetradecamer GroEL_14_ particle: (**a**) SDS-PAGE electrophoresis of the GroEL_14_-EGFP population (Lane 2). The intensity of the WT GroEL subunit band is 9.69 relative units while that of the fusion GroEL_1_-EGFP is 1. Molecular-weight markers (Lane 1). (**b**) Absorption spectra of the GroEL_14_-EGFP population (black solid line), monomeric fusion GroEL_1_-EGFP (black dashed line, absorbance at 489 nm is A_489nm_ = 0.06 absorbance units (AU), concentration of the protein is C = 1.07 µM). The absorption spectrum of WT GroEL_14_ subunits (black dashed-dot line) was obtained by subtraction of the GroEL_1_-EGFP spectrum from the spectrum of the GroEL_14_-EGFP population after their normalization at 499 nm (EGFP absorption) with a negligible contribution of the light scattering, and by the correction for light scattering, A_280nm_ = 0.12 AU, C = 14.3 µM.

**Figure 3 molecules-28-01901-f003:**
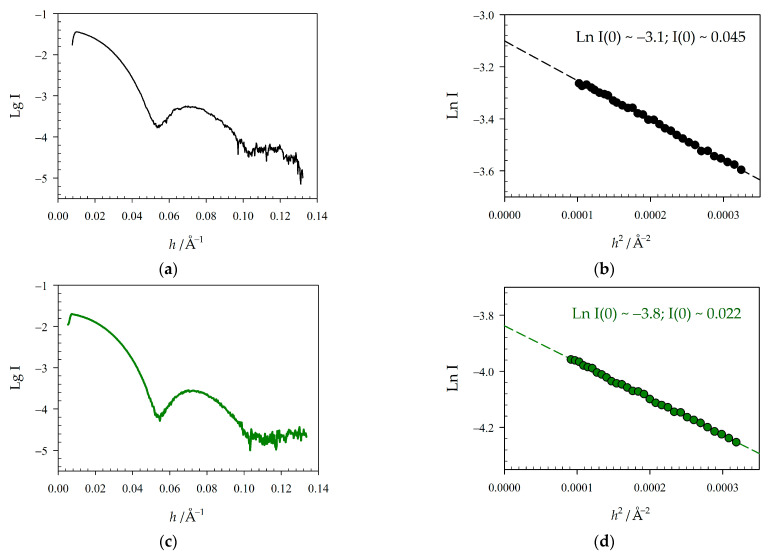
SAXS pattern of WT GroEL_14_ population of 5 mg/mL (**a**, indicated in black) and GroEL_14_-EGFP population of 2.5 mg/mL (**c**, indicated in green); (**b**,**d**) corresponding Guinier plots; scattering vector *h* = 4×(π/λ) × sin(θ), where 2θ—scattering angle and λ—wavelength; I—X-ray intensity. Radius gyration Rg = 67.7 ± 0.2 (Å) for WT GroEL_14_ and 62.5 ± 0.2 (Å) for GroEL_14_-EGFP. Buffer: 20 mM Tris-HCl pH 7.6, 100 mM KCl, 10 mM MgCl_2_.

**Figure 4 molecules-28-01901-f004:**
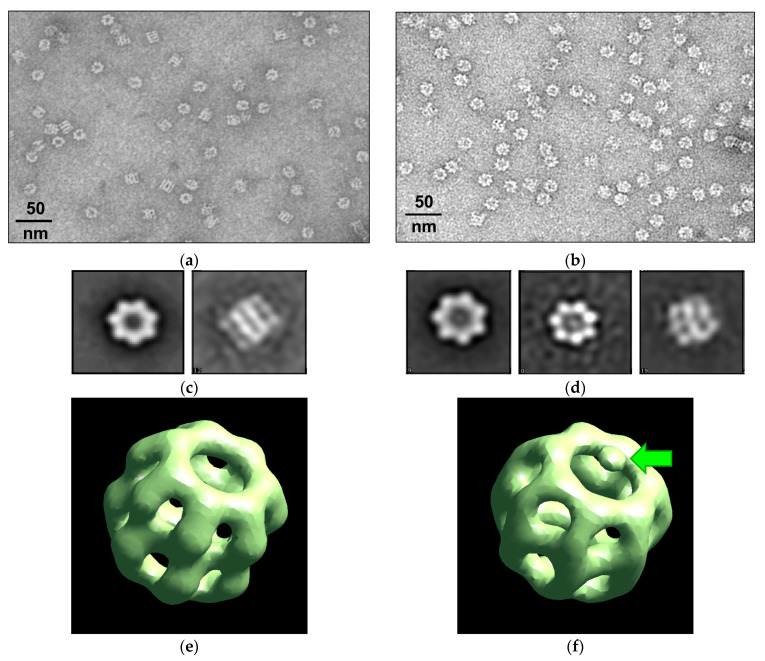
The reconstruction of the GroEL_14_ and GroEL_14_-EGFP spatial structures using their TEM images: the field and identified averaged top and side views of GroEL_14_ (**a**,**c**) and GroEL_14_-EGFP (**b**,**d**); the respective reconstructions (**e**,**f**). The green arrow indicates the center of the EGFP position in the reconstruction.

**Figure 5 molecules-28-01901-f005:**
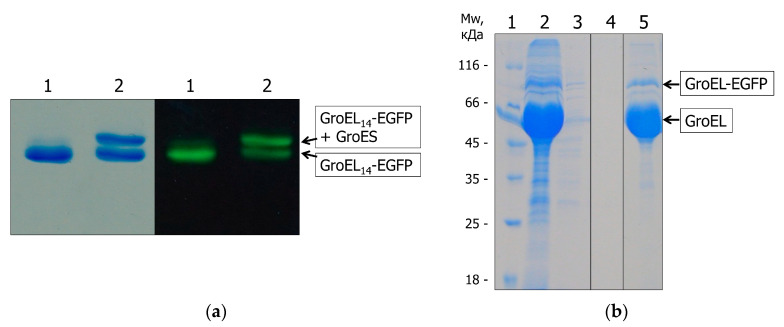
Binding of purified GroEL_14_-EGFP to GroES (**a**); binding of GroEL_14_-EGFP from cell extract (coexpression, see Materials and Methods (Section 4)) to denatured lysozyme (**b**). (**a**) Nondenaturing PAGE in the presence of Mg-ADP without GroES (Lane 1) and after incubation with GroES at the 1:5 molar ratio (Lane 2), as detected by Coomassie staining (left panel) and EGFP fluorescence (right panel) in the same electrophoretic gel; (**b**) SDS-PAGE of the results of affinity chromatography based on denatured lysozyme of the cell extract (coexpression): Lane 1, molecular-weight markers; Lane 2, loading; Lane 3, breakthrough; Lane 4, washing; Lane 5, elution by 6 M urea. Three lanes are three parts of the same gel.

**Figure 6 molecules-28-01901-f006:**
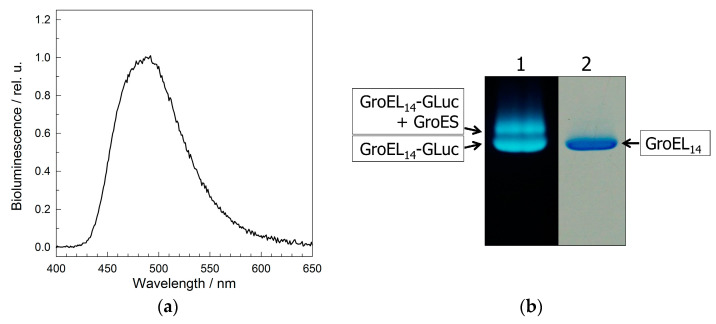
Physicochemical properties of GroEL_14_ with incorporated *Gaussia princeps* luciferase (Gro-EL_14_-GLuc): (**a**) the bioluminescence spectrum of cells after coexpression of pAC28 groEL-Luc and pET11c groES/EL; (**b**) nondenaturing PAGE of the cell lysate in the presence of Mg-ADP (Lane 1) and purified GroEL_14_ (Lane 2). Protein band visualization by bioluminescence (Lane 1) and Coomassie staining (Lane 2).

**Figure 7 molecules-28-01901-f007:**
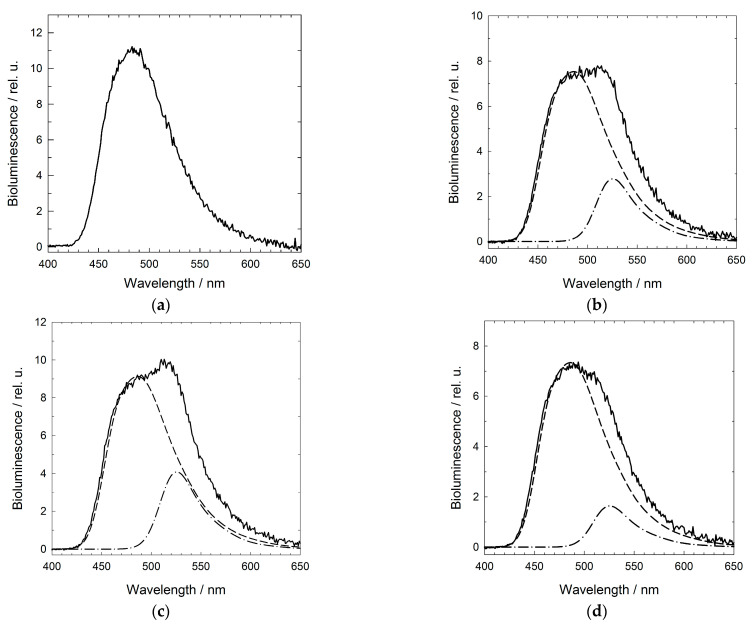
The interaction of purified GroEL_14_-GLuc with fluorescein-labeled denatured proteins monitored by BRET: the bioluminescence spectra of GroEL_14_-GLuc in the absence of denatured proteins (**a**) and in the presence of denatured lysozyme (**b**), α-lactalbumin (**c**), and malate dehydrogenase (**d**). The dashed lines show the contribution of bioluminescence to the spectrum; the dashed-dot lines are the contribution of BRET-induced fluorescein fluorescence.

**Figure 8 molecules-28-01901-f008:**
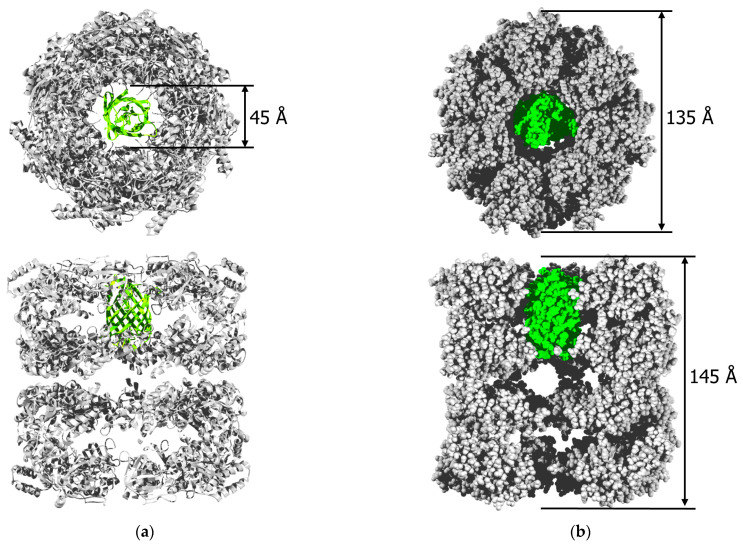
A model of GroEL_14_ with EGFP incorporated into the inner cavity of one of the protein rings. (**a**) Ribbon view from top and side with three front subunits deleted; (**b**) space-fill view from top and side with three front subunits deleted. GroEL is in gray, and EGFP is in green. Structures used in modeling are from PDB ID 1OEL for GroEL [38] and PDB ID 2Y0G for EGFP [39]; the presentation was composed using SPDBViewer 4.1.0 software (developed by Nicolas Guex, Geneva, Switzerland) [40], and POV-Ray 3.7.0 software (Persistence of Vision Raytracer Pty. Ltd., Williamstown, Victoria, Australia).

## Data Availability

Not applicable.

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
