# Peer review of "In Vivo Incorporation of Photoproteins into GroEL Chaperonin Retaining Major Structural and Functional Properties"

_molecules, 2023, doi:10.3390/molecules28041901_

Round 1

Reviewer 1 Report

The paper "In vivo Incorporation of Photoproteins into GroEL Chaperonin Retaining Major Structural and Functional Properties" describes a strategy to produce GroEL chaperonin bound to two photoproteins. Beside to the production of the macromolecular system, physico-chemical and structural techniques have been used to characterize the produced sample. This referee considers the paper well written and with a topic that fits to the journal "Molecules".

Below, few points that could make more clear the paper:

1) In the paper, a description of previous works reporting similar incorporations in chaperonine systems lacks and has to be included in the introduction part;

2) The authors exploit fluorescence or bioluminescence to prove the correct folding of the incorporated photoprotein (see lines 101-103 for GroEL14-EGFP and 188-189 for GroEL14-Gluc). However, such analyses only allow assessing that the folding assumed by the protein produce fluorescence or bioluminescence and not that such conformation is the correct one (I suppose that correct is used instead of physiological). Moreover, it could be possible that the signal comes from a mix of protein "correctly" folded and unfolded because a ratio between signal respect to the expected one in the case all protein produces florescence signal has not been assessed. Therefore, the sentence at lines 101-102 and 188-189 has to be changed accordingly.

3) between 117 and 125 lines, electrophoresis and UV-Vis measurements have been reported about the protein resulting from the co-expression with pAC28 groEL-EGFP and pET11c groEL. Such measurements show the ratio between GroEL-EGFP and GroEL14 and not that GroEL-EGFP is incorporated in one of the two rings of GroEL14 (information that, instead, results from structural technique such as CryoEM which shows the electron density of the chaperonin and of the photoprotein). Therefore, the sentence at line 124-125 has to be changed accordingly.

4) information about Guinier analysis such as the sRg limit used and I(0) (that could be informative for the MW of the samples) are not included in the paragraph 2.2. Similarly, resolution of CryoEM structure has not been reported. Please, add such information.

5) Information about data analysis, software for data integration and Guinier analysis lacks in the material and methods paragraph 4.6. Please, add such information

Author Response

Dear Reviewer,

Thank you for the rigorous reading of our manuscript and valuable remarks to improve the text and data presentation. We accepted all your considerations and changed the text according to them.

Point 1: In the paper, a description of previous works reporting similar incorporations in chaperoninesystems lacks and has to be included in the introduction part.

Response 1: Thank you for pointing out the ambiguous place. Despite previous works reporting similar
incorporations of alien polypeptides in chaperonin of various origins being very limited, we included some most appropriate examples in the Introduction (lines 53-87).

Point 2: The authors exploit fluorescence or bioluminescence to prove the correct folding of the incorporated photoprotein (see lines 101-103 for GroEL14-EGFP and 188-189 for GroEL14-Gluc). However, such analyses only allow assessing that the folding assumed by the protein produce fluorescence or bioluminescence and not that such conformation is the correct one (I suppose that correct is used instead of physiological). Moreover, it could be possible that the signal comes from a mix of protein "correctly" folded and unfolded because a ratio between signal respect to the expected one in the case all protein produces florescence signal has not been assessed. Therefore, the sentence at lines 101-102 and 188-189 has to be changed accordingly.

Response 2: Thank you very much for the correction. We changed the sense in corresponding sentences in lines 140 and 264.

Point 3: between 117 and 125 lines, electrophoresis and UV-Vis measurements have been reported about the protein resulting from the co-expression with pAC28 groEL-EGFP and pET11c groEL. Such measurements show the ratio between GroEL-EGFP and GroEL14 and not that GroEL-EGFP is incorporated in one of the two rings of GroEL14 (information that, instead, results from structural technique such as CryoEM which shows the electron density of the chaperonin and of the photoprotein). Therefore, the sentence at line 124-125 has to be changed accordingly.

Response 3: Thank you very much for the correction. We removed the sentence in lines 166-167 accordingly to your remark.

Point 4: information about Guinier analysis such as the sRg limit used and I(0) (that could be informative for the MW of the samples) are not included in the paragraph 2.2. Similarly, resolution of CryoEM structure has not been reported. Please, add such information.

Response 4: Thank you very much for the remarks and conclusions. Due to these remarks, we more carefully analyzed the SAXS data available in stock and found inaccuracies in the presentation of Figure 3. Besides, we found the publication of Piiadov et al. that allows us independently check the parameters of the Guinier plots and estimate the molecular weight of proteins using only the values of their radius of gyration. We expanded paragraphs 2.2, 4.6, and 4.7 by including some information about the Rg and I(0) determination from SAXS experiments and the details of the structural reconstruction of ETM images
(lines 185-203, 205-209, 228-230, 467-469, 482-487, and 490-492).

Point 5: Information about data analysis, software for data integration and Guinier analysis lacks in the material and methods paragraph 4.6. Please, add such information.

Response 5: Thank you very much for the correction. We added in paragraph 4.6 the information about software for SAXS integration and Guinier analysis (lines 467-469).

Sincerely yours,
Gennady V. Semisotnov,
Corresponding author

Reviewer 2 Report

The research is well-performed and fun to read, though the results' significance is relatively low.

A wide array of control measurements was performed and carefully interpreted. As a result, I have high confidence in the validity of the presented results. However, what I missed is a Justification or outlook on what this kind of technique could be used for. For now, the main take-home message seems to be "look what you can do with GRoEl!".

Author Response

Dear Reviewer,
Thank you for your careful analysis and good evaluation of our work. Although one could agree with you that the significance of the results is relatively low, it seems to me that the approach proposed in the work to incorporate other proteins in oligomeric proteins will find its performers.

Point 1: The research is well-performed and fun to read, though the results' significance is relatively low.
A wide array of control measurements was performed and carefully interpreted. As a result, I have high confidence in the validity of the presented results. However, what I missed is a Justification or outlook on what this kind of technique could be used for. For now, the main take-home message seems to be "look what you can do with GRoEl!".

Response 1: Thank you for the valuable comment and pointing out the ambiguous place. We have added a section to the Introduction which helps better understand the application area of methods for introducing polypeptides into GroEL-like chaperonins (lines 53-87).

Sincerely yours,
Gennady V. Semisotnov,
Corresponding author